# Text-Guided Diffusion Image Style Transfer with Contrastive Loss

## Abstract

Recently, diffusion models have demonstrated superior performance in text-guided image style transfer. However, due to the stochastic nature of the diffusion models, there exists a fundamental trade-off between transforming styles and maintaining content in the diffusion models. Although a simple remedy would be using deterministic sampling schemes such as denoising diffusion implicit model (DDIM) that guarantees the perfect reconstruction, it requires the computationally expensive fine-tuning of the diffusion models. To address this, here we present a text-guided sampling scheme using a patch-wise contrastive loss. By exploiting the contrastive loss between the samples and the original images, our diffusion model can generate an image with the same semantic content as the source image. Experimental results demonstrate that our approach outperforms the existing methods while maintaining content and requiring no additional training on the diffusion model.

## 1 Introduction

Style transfer is the task that converts the style of a given image into another style while preserving its content. Over the past few years, GAN-based methods such as pix2pix (Isola et al., 2017), cycleGAN (Zhu et al., 2017), and contrastive unpaired image-to-image translation (CUT) have been developed (Park et al., 2020). Recently, joint use of a pretrained image generator and image-text encoder enabled text-guided image editing which requires little or no training of the networks (Radford et al., 2021; Crowson et al., 2022; Patashnik et al., 2021; Gal et al., 2022; Kwon & Ye, 2022).

Inspired by the success of diffusion models for image generation (Ho et al., 2020; Song et al., 2020), image editing (Liu et al., 2021), in-painting (Avrahami et al., 2022), super-resolution (Chung et al., 2022), etc., many researchers have recently investigated the application of the diffusion models for image-to-image style transfer (Saharia et al., 2022; Su et al., 2022). For example, (Saharia et al., 2022; 2021) proposed conditional diffusion models that require paired dataset for image-to-image style transfer. One of the limitations of these approaches is that the diffusion models need to be trained with paired data set with matched source and target styles. As collecting the matched source and target domain data is impractical, many recent researchers have focused on unconditional diffusion models.

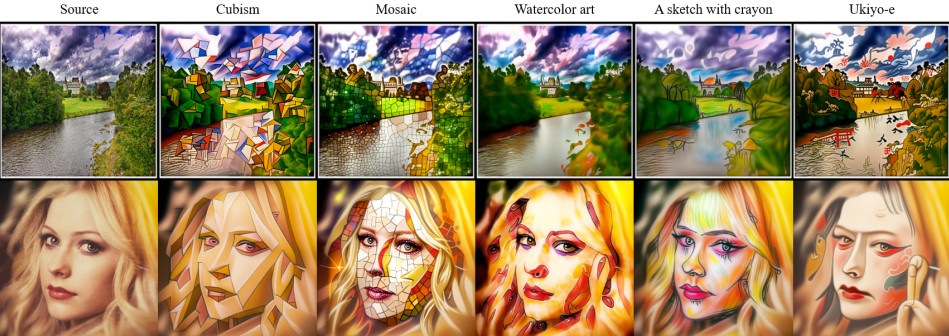

Figure 1: Results of our style transfer method on various artistic styles. The source images are translated into various styles while maintaining their structure.

For example, the dual diffusion implicit bridge (DDIB) (Su et al., 2022) exploits two score functions that have been independently trained on two different domains. Although DDIB can translate one image into another without any external condition, it also requires training of two diffusion models for each domain which involves additional training time and a large amount of dataset. On the other hand, DiffusionCLIP (Kim et al., 2022) leverages the pretrained diffusion models and CLIP encoder to enable text-driven image style transfer without additional large training data set. Unfortunately, DiffusionCLIP still requires additional fine-tuning of the model for the desired style. Besides the additional complexity, unconditional diffusion models for image style transfer have further limitations in maintaining content. This is because the reverse sampling procedure of the diffusion models does not have an explicit constraint to impose the content consistency, and the stochastic nature of diffusion models makes them easy to change the content and styles at the same time.

To address this, here we propose a diffusion model that transfers the style of a given image while preserving its semantic content by using contrastive loss similar to CUT (Park et al., 2020). Since contrastive loss can exploit the spatial information in terms of positive and negative pairs, we found that diffusion model already contains the spatial information that can be used to maintain the content. Furthermore, in contrast to DiffusionCLIP, it only requires the CUT loss fine-tuning via light weighted multi-layer perceptron (MLP) layers rather than the diffusion model, so the computational complexity can be significantly reduced. Even more, thanks to the extracted spatial features from diffusion models, we observe that the MLP fine-tuning is not even necessary with slight decrease in quality.

To verify the effectiveness of this method, we present a text-driven style transfer using CLIP (Radford et al., 2021). In particular, we utilize CLIP in a patch-wise manner similar to (Kwon & Ye, 2022) thanks to its stable style translation. Our contribution can be summarized as following:

- Thanks to the content disentanglement using contrastive loss, to our best knowledge, our method is the first style transfer method with unconditional diffusion model that overcomes the trade-off between style and content.

- Our method only requires contrastive loss from the pre-trained diffusion models rather than fine-tuning the diffusion model for target domain, so the computational complexity is much low but still allows effective image transfer to any unseen domain.

## 2 RELATED WORKS

**Image style transfer**   Neural style transfer (Gatys et al., 2016) is the first approach to change the style texture of the content image into a style image by iterative optimization process. However, these iterative process takes significant amount of time. Alternatively, the adaptive instance normalization (AdaIN) by (Huang & Belongie, 2017) converts the means and variances of the features of the source image to those of the target image, which enables arbitrary style transfer.

On the other hand, pix2pix (Isola et al., 2017), CycleGAN (Zhu et al., 2017) and CUT (Park et al., 2020) rely on different mechanisms for content preservation. Specifically, in CycleGAN (Zhu et al., 2017), the cycle consistency assumes bijective relationship between two domains for content preservation, whose constraint is often restrictive in some applications. In order to overcome this restriction, CUT (Park et al., 2020) was proposed to maximize the mutual information between the content input and stylized output images in a patch-based manner on the feature space. This leads to preservation of the structure between the two images while changing appearance.

With the advent of CLIP model (Radford et al., 2021), it has been shown that text-guided image synthesis can be accomplished without collecting style images. CLIP has semantic representative power which results from large scale dataset consisting of 400 millions image and text pairs. This enables text-driven image manipulation. StyleCLIP (Patashnik et al., 2021) was proposed to optimize latent vector of the content input given text prompt by using CLIP and pretrained StyleGAN (Karras et al., 2020). However, image modification using StyleCLIP is limited to the domain of the pretrained generator. In order to solve this issue, StyleGAN-NADA (Gal et al., 2022) presented out-of-domain image manipulation method that shifts the generative model to new domains. VQGAN-CLIP (Crowson et al., 2022) has shown that VQGAN (Esser et al., 2021) can also be used as a pretrained generative model to generate or edit high quality images without training. In order not to be restricted to the domains of the pretrained generators, CLIPstyler (Kwon & Ye, 2022) proposed a

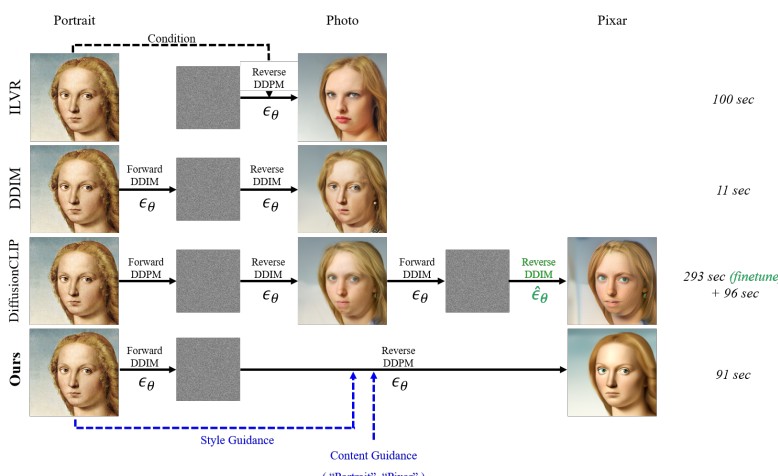

Figure 2: An illustration on sampling schemes of four diffusion models for style transfer.

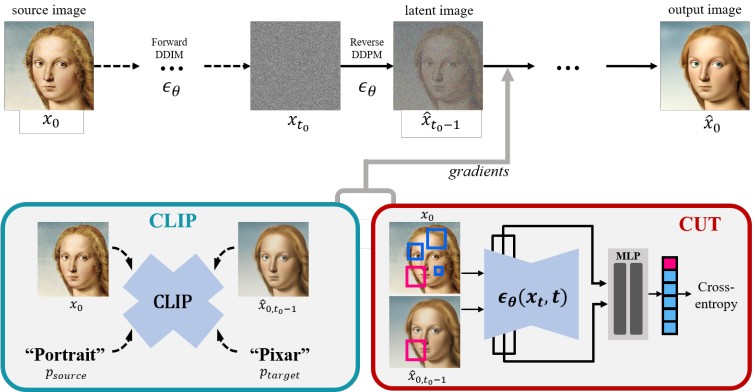

Figure 3: Our proposed method. In order to guide the diffusion model, CUT loss is calculated using the noise estimator $\epsilon_\theta(\cdot)$ and CLIP loss using the CLIP model. Through these losses, gradients are added to the predicted mean at each time step. More details regarding CUT implementation on the diffusion model are described in METHOD and APPENDIX sections.

CNN encoder-decoder model that learns both content and style properties through patch-wise CLIP loss.

**Diffusion models for style transfer**   Diffusion probabilistic models have attracted great attention because of their superior performance in generating images despite their long training time (Ho et al., 2020; Song et al., 2020). The diffusion model is a generative model that involves Markov chain of forward process by gradually adding Gaussian noise (Ho et al., 2020). Then, with a trained noise estimation model, clean samples are generated from the latent noise by iterative denoising process. They have been applied to a wide variety of computer vision areas involving super resolution (Rombach et al., 2022), segmentation (Baranchuk et al., 2022), image editing (Avrahami et al., 2022), medical image processing (Kim et al., 2021), and video generation (Ho et al., 2022).

Recently, there have been several attempts to translate images from one domain to another using diffusion models. Palette (Saharia et al., 2022) demonstrated that various types of image-to-image translation can be performed by utilizing conditional diffusion models which require paired dataset for training. For unconditional diffusion models, DDIB (Su et al., 2022) proposed to exploit two score functions which have been independently trained on two different domains. However, it requires training diffusion models on each style domain, which takes a significant amount of time. Meanwhile, DiffusionCLIP (Kim et al., 2022) was proposed for image manipulation using a pretrained diffusion model. Specifically, DiffusionCLIP fine-tunes the model with identity and style losses, which explicitly impose constraints on transferring appearance while maintaining structure.

Though it could be trained easily with semantic knowledge of CLIP, the model needs to be fine-tuned for each style domain. Additionally, diffusion models with guidance from regression models have been proposed for image-to-image translation tasks (Wolleb et al., 2022). Although it could transform the content images with the desired style without additional training on diffusion models, its generated image suffers from inconsistency with the content image.

# 3 METHODS

## 3.1 DILEMMA OF DIFFUSION MODELS

For image style transfer, the generative process should convert an image into a given style while retaining the content. However, diffusion models have difficulties in maintaining the semantic information. Specifically, in diffusion models, given a source data distribution $q(x_0)$, latent variable $x_t$ is computed by forward diffusion process. DDPM (Ho et al., 2020) directly samples $x_t$ from $x_0$ by adding Gaussian noise with $\beta_t \in (0, 1)$ at time $t \in [1, ..., T]$,

$$x_t = \sqrt{\overline{\alpha}_t}x_0 + \sqrt{1 - \overline{\alpha}_t}\epsilon \tag{1}$$

where $\epsilon \sim \mathcal{N}(0, \boldsymbol{I})$, $\alpha_t = 1 - \beta_t$, and $\overline{\alpha}_t = \prod_{i=0}^{t} \alpha_i$. The reverse sampling process to generate a clean image is then given by:

$$x_{t-1} = \frac{1}{\sqrt{1 - \beta_t}} \left( x_t - \frac{\beta_t}{\sqrt{1 - \overline{\alpha}_t}}\epsilon_\theta(x_t, t) \right) + \sigma_t\epsilon. \tag{2}$$

Although the noise $\epsilon$ contributes to achieve sample diversity, it in return leads to a loss of content in the context of style transfer. The stack of these stochastic steps can result in the images with completely different content even if the images are sampled from the same intermediate latent. In order to preserve the semantics, ILVR (Choi et al., 2021) in Fig. 2 tried to generate diverse samples with image condition, but still suffers from the stochasticity by $\epsilon$. Meanwhile, the content can be preserved with DDIM (Song et al., 2020) whose sampling process is:

$$x_{t-1} = \sqrt{\overline{\alpha}_{t-1}}f_\theta(x_t, t) + \sqrt{1 - \overline{\alpha}_{t-1} - \sigma_t^2}\epsilon_\theta(x_t, t) + \sigma_t^2\epsilon \tag{3}$$

where $\sigma_t$ is the variance of noise which controls how stochastic the sampling process is, and $f_\theta$ is given by:

$$f_\theta(x_t, t) := \frac{x_t - \sqrt{1 - \overline{\alpha}_t}\epsilon_\theta(x_t, t)}{\sqrt{\overline{\alpha}_t}}. \tag{4}$$

When $\sigma_t = 0$, the noise term $\epsilon$ is removed from (3) and then we can successfully preserve the content. Since the sampling process is deterministic, however, the style is also preserved, which is not desired for style transfer as shown in Fig. 2. DiffusionCLIP in Fig. 2 tried to overcome this problem by fine-tuning $\epsilon_\theta$, which takes much time and computation. In order to solve this dilemma, we propose diffusion-based style transfer method using guidance that requires no additional training on the diffusion model and is applicable even to unseen domains.

## 3.2 GUIDANCE FOR DIFFUSION MODELS

Guiding gradients in diffusion models is the method proposed in the context of class-conditional image generation (Dhariwal & Nichol, 2021). Accordingly, even the unconditional diffusion model can generate conditional images using guidance by classifiers or CLIP.

Specifically, as proposed in the past work (Avrahami et al., 2022), gradients for the guidance is calculated as

$$\nabla_{\hat{x}_{0,t}} L_{total}(\hat{x}_{0,t}, x_0, p_{target}, p_{source})$$

where $p_{target}$ and $p_{source}$ are text prompts for target and source domain, respectively. From DDPM, the estimated clean image $\hat{x}_{0,t}$ from the latent variable $x_t$ (Avrahami et al., 2022) can be produced from the noise approximation model $\epsilon_\theta(x_t, t)$:

$$\hat{x}_{0,t} = \frac{x_t}{\sqrt{\overline{\alpha}_t}} - \frac{\sqrt{1 - \overline{\alpha}_t}\epsilon_\theta(x_t, t)}{\sqrt{\overline{\alpha}_t}}. \tag{5}$$

The gradients are then added to the predicted mean at each time step $t$. The guiding method is illustrated in Figure 3. In order to achieve superior performance, we incorporate two types of guidance - style and content guidance.

**CLIP loss for style guidance**  The CLIP model is trained on extensive language and image dataset which results in its great semantic power (Radford et al., 2021). Thanks to this semantic capacity, we can generate images in diverse styles with only text prompts. The CLIP loss for style guidance can be formulated as:

$$\mathcal{L}_{CLIP} = \mathcal{L}_{global}(\hat{x}_{0,t}, p_{target}) + \mathcal{L}_{directional}(\hat{x}_{0,t}, x_0, p_{target}, p_{source}) \qquad (6)$$

The global CLIP loss $L_{global}$ calculates the cosine distance in the CLIP embedding space between the generated image $\hat{x}_{0,t}$ and the style prompt $p_{target}$ (Patashnik et al., 2021) as follows:

$$\mathcal{L}_{global}(\hat{x}_{0,t}, p_{target}) = D_{CLIP}(\hat{x}_{0,t}, p_{target}). \qquad (7)$$

Since the global loss suffers from mode collapse and corrupted image quality, the directional CLIP loss $L_{directional}$ was proposed (Gal et al., 2022). It aligns the direction in the CLIP embedding space between text and image pairs. In our case, it can be formulated as:

$$\mathcal{L}_{directional}(\hat{x}_{0,t}, x_0, p_{target}, p_{source}) = 1 - \frac{\Delta I \cdot \Delta T}{\parallel \Delta I \parallel \parallel \Delta T \parallel} \qquad (8)$$

where $\Delta I = E_{img}(x_0) - E_{img}(\hat{x}_{0,t})$, $\Delta T = E_{txt}(p_{source}) - E_{txt}(p_{target})$ for CLIP's image encoder $E_{img}$ and text encoder $E_{txt}$. Meanwhile, the patch-based CLIP loss was proposed to enhance the generated images' quality (Kwon & Ye, 2022). So, we adopt the patch-based scheme in both $L_{global}$ and $L_{directional}$.

**CUT loss for content guidance**  With respect to style transfer task, it is essential that diffusion models not only transform the style, but also preserve the content. CUT loss (Park et al., 2020) has been proven to effectively maintain structure information by maximizing mutual information between corresponding input and output patches. It requires training an encoder which can capture the spatial information from the input. Features $z$ extracted from the encoder are then used for contrastive learning. Meanwhile, the noise predictor, U-net in the diffusion model, has been shown to contain spatial information (Baranchuk et al., 2022). Thus, we could easily apply the patch-wise contrastive loss in order to preserve contents by utilizing the spatial information extracted from the diffusion model without its additional training as shown in Figure 3. It can be formulated as follows:

$$\mathcal{L}_{CUT}(\hat{x}_{0,t}, x_0) = \mathbb{E}_{x_0} \left[ \sum_l \sum_s \ell(\hat{z}_\ell^s, z_\ell^s, z_\ell^{S \setminus s}) \right] \qquad (9)$$

where $\hat{z}_l$ and $z_l$ are $\epsilon_\theta$'s $l$-th layer features from $\hat{x}_{0,t}$ and $x_0$, respectively. We denote $s$ as a spatial location in $\{1, \ldots, S_l\}$ where $S_l$ is the number of spatial locations in the feature $z_l$. Additionally, $\ell(\cdot)$ is cross-entropy loss. By minimizing the CUT loss, we can maintain the semantic consistency between the reverse sampled image $\hat{x}_{0,t}$ and the original image $x_0$ so that the content can be preserved. More details are described in the Section A.2.

It is important how contrastive loss for content guidance is incorporated into the diffusion model. We exploit two methods. First, as illustrated in the Figure 3, feature maps are extracted from the encoder part of the noise estimator $\epsilon_{\theta,enc}(x_t, t)$. Then, the feature maps are forwarded to the MLP network $F$. Then, the feature maps $z_i$'s for the CUT loss are extracted as the output of the MLP network $F$. The main motivation of using MLP network $F$ is to fine-tune the network so that better spatial features are extracted for contrastive loss. In the second method, we bypass the MLP network and extract features directly from the diffusion model so that we can further reduce the computation time and complexity of the MLP network with a slight loss of image quality.

For the case of MLP fine-tuning, the network $F$ consists of two linear layers and ReLU activation between them. The number of output channels are 256. $L2$ norm of the outputs become the final outputs, which are then utilized to calculate cross entropy loss. The outputs can be written as:

$$z_l = F_l(\epsilon_{\theta,enc}(x_0)), \quad \hat{z}_l = F_l(\epsilon_{\theta,enc}(\hat{x}_0)). \qquad (10)$$

In order to fine-tune the network $F$, we carry out two stages, each consisting of forward and reverse processes. During these two stages, $\epsilon_\theta(x_t, t)$ is fixed. At the first stage, the network $F$ is fine-tuned.

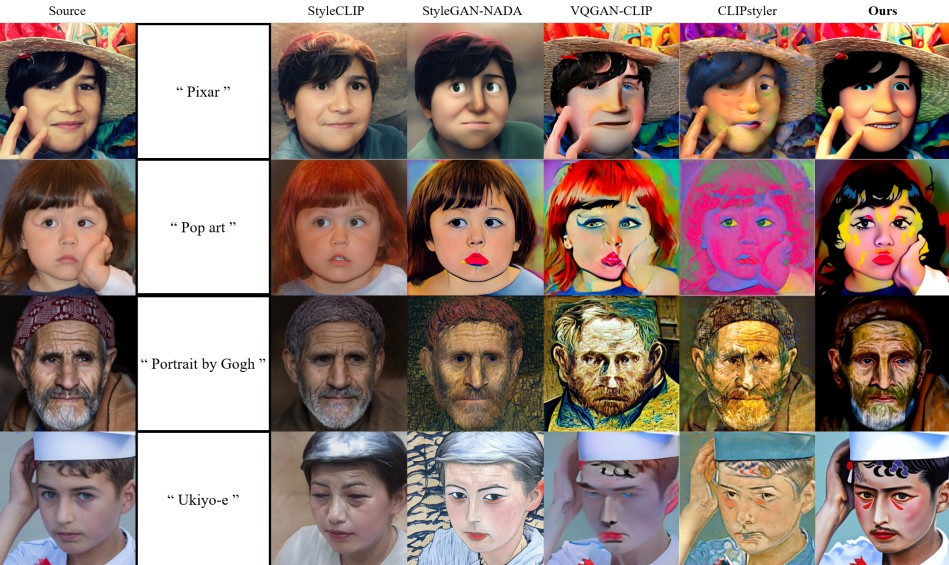

Figure 4: Comparison against GAN-based style transfer methods. Our method outperforms four GAN-based methods in perspective of both style transformation and content preservation.

| Methods | User study | | CLIP score ↑ | Face ID ↓ |
|---|---|---|---|---|
| | Content ↑ | Style ↑ | | |
| StyleCLIP | 3.84 | 1.46 | 0.0925 | **0.3750** |
| StyleGAN-NADA | 3.15 | 2.71 | 0.1222 | 0.4948 |
| VQGAN-CLIP | 1.83 | 2.79 | 0.1379 | 0.7661 |
| CLIPstyler | 2.05 | 2.93 | 0.1347 | 0.6664 |
| Ours | **4.33** | **4.39** | **0.1483** | 0.4219 |

Table 1: User study and quantitative results on GAN-based methods for style transfer. The bold text and underline refer to the best and second best results, respectively.

From the source image $x_0$, we can get $x_{t_0}$ from the forward noising process, where $t_0 \in [0, T]$. As shown in the Figure 3, we can reverse $x_{t_0}$ to $\hat{x}_{t_0-1}$ with other style and content guidance excluding CUT loss. From (5), $\hat{x}_{0,t_0-1}$ can be obtained. With $\hat{x}_{0,t_0-1}$ and $x_0$, we can obtain feature maps from $\epsilon_\theta(x_t, t)$ and fine-tune the network $F$ using the CUT loss. Again, we can reverse $\hat{x}_{t_0-1}$ with guidance in order to get $\hat{x}_{t_0-2}$. By repeating these procedures, we can get $\hat{x}_0$. Then, MLP fine-tuning is finished. At the second stage, all the networks, including $F$, are fixed and the outputs are generated with style and content guidance including CUT loss.

**Total loss** On top of the contrastive loss, we include the feature loss $L_{VGG}$, which is the mean-squared error between the VGG feature maps of $\hat{x}_{0,t}$ and $x_0$, and the pixel loss $L_{MSE}$, which is the $L_2$ norm of the pixel difference between them.

$$\mathcal{L}_{content} = \mathcal{L}_{CUT}(\hat{x}_{0,t}, x_0) + \mathcal{L}_{VGG}(\hat{x}_{0,t}, x_0) + \mathcal{L}_{MSE}(\hat{x}_{0,t}, x_0) \quad (11)$$

Therefore, the total loss function for the guidance is formulated as:

$$\mathcal{L}_{total} = \mathcal{L}_{CLIP}(\hat{x}_{0,t}, x_0, p_{target}, p_{source}) + \mathcal{L}_{content}(\hat{x}_{0,t}, x_0 \quad (12)$$

The weights for each loss function are hyperparameters which need to be chosen by users. The examples of these weights are given in the Section A.1 and Table 6.

## 4 EXPERIMENTS

### 4.1 EXPERIMENTAL SETTING

**Dataset** The images used as content reference are from FFHQ (Karras et al., 2019), CelebA-HQ (Karras et al., 2017), ImageNET (Deng et al., 2009), LSUN-Church (Yu et al., 2015), and CycleGAN dataset (Zhu et al., 2017). They contain images of human faces, objects, scenes, and churches.

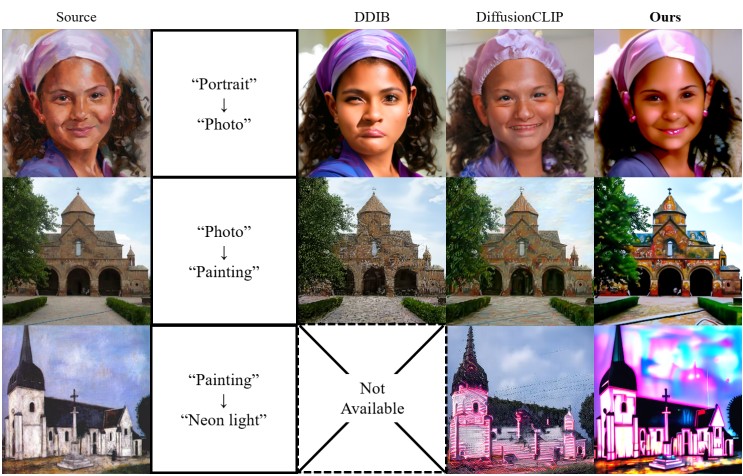

Figure 5: Comparison between three diffusion-based style transfer methods. Our proposed method can modulate styles from unseen domain images that other diffusion models cannot.

| Methods | Photo domain | | Unseen domain | |
|---|---|---|---|---|
| | Content ↑ | Style ↑ | Content ↑ | Style ↑ |
| DiffusionCLIP | 4.05 | 2.90 | 3.20 | 2.75 |
| Ours | **4.70** | **4.70** | **4.55** | **4.55** |

Table 2: User study results on comparison against DiffusionCLIP.

Furthermore, in order to evaluate the performance of our proposed model on the images from unseen domains, we utilize Wikiart dataset (Danielczuk et al., 2019). All the images are resized to $256 \times 256$ for the diffusion models. For patch-based guidance, we randomly crop 96 patches from a source image and then apply perspective augmentation and affine transformation. More details are illustrated in the Appendix.

**Pretrained diffusion models** We utilize the unconditional diffusion model trained on ImageNET dataset with $256 \times 256$ image size (Dhariwal & Nichol, 2021) and the model trained on FFHQ dataset with $256 \times 256$ image size (Choi et al., 2021).

**Sampling scheme** Either DDIM or DDPM method can be applied in our method during the forward and reverse diffusion steps. We basically adopt the DDIM strategy as the forward noising process and DDPM method as the reverse sampling. When $T$ is the total time step, we respace the step size from $T$ to $T'$. Then with the source image $x_0$, we obtain the latent $x_{t_0}$ from the forward diffusion process, where $t_0 \in [0, T']$. We choose $(T', t_0)$ as $(50, 25)$ as default when $T = 1000$. From this latent $x_{t_0}$, the stylized output image is sampled through diffusion processes. In this way, not only can more latent information be preserved from the source image, but the image can be sufficiently converted to a new style at the same time. By greatly reducing the number of iterations, inference time could be significantly reduced. The sampling scheme is illustrated in Figs. 2, 3 and the comparative studies on the choice of $(T', t_0)$ are illustrated in the Section B.1.

### 4.2 COMPARATIVE STUDIES

Figure 1 shows that our method achieves outstanding results on various artistic styles. In addition, we perform comparisons with GAN-based and diffusion-based style transfer methods, respectively.

#### 4.2.1 GAN-BASED MODELS

For GAN-based models, we compare four state-of-the-art methods - StyleCLIP (Patashnik et al., 2021), StyleGAN-NADA (Gal et al., 2022), VQGAN-CLIP (Crowson et al., 2022), CLIPstyler (Kwon & Ye, 2022). The results of the comparison are illustrated in Figure 4. We could clearly see that our proposed model outperforms in aspect of retaining the content. The generated outputs from both StyleCLIP and StyleGAN-NADA show distorted results that non-face objects, such as hands or hats, are removed from the output images. Although results from VQGAN-CLIP and CLIPstyler show relatively better result in perspective of content, we observe that some details of the face, such as eyes or mouth, are crumbled. In contrast, our proposed method does not compromise

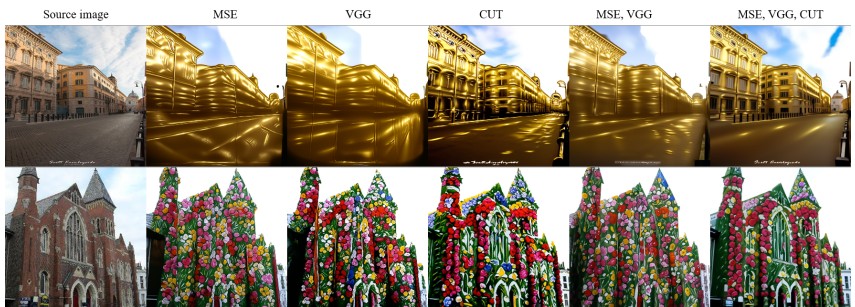

Figure 6: Ablation study on three losses for content guidance - $L_{MSE}$, $L_{VGG}$, $L_{CUT}$. In each row, source images are translated into the style of "golden" and "oil painting of flowers", respectively. Patch-wise contrastive loss helps to preserve content information.

|  | MSE | VGG | CUT | MSE, VGG | MSE, VGG, CUT |
|---|---|---|---|---|---|
| **Content** ↑ | 2.80 | 2.05 | 3.90 | 2.90 | **4.75** |
| **Style** ↑ | 2.05 | 3.10 | 3.85 | 3.70 | **4.20** |

Table 3: User study results on ablation studies regarding content losses. Bold text and underline refer to the best and the second best scores, respectively.

the structural information in that hats and hands are retained in the outputs, and neither hairs nor eyes are crushed. Also, the results show that our proposed method generates outputs with feasible texture. StyleCLIP gives unsatisfactory results that still look like photos. StyleGAN-NADA has difficulty in translating images into pop art style. VQGAN-CLIP and CLIPstyler also fail at generating Pixar and Uiyo-e style images. On the contrary, we could see that our method provides high fidelity samples which are transferred into the styles of the target prompt.

In addition to the qualitative results, the superiority of our method is verified by quantitative results. In Table 1, we could see that our proposed method achieves the best scores in both user study and CLIP score. Although StyleCLIP attains the smallest loss in face identity, this implies that StyleCLIP maintains semantic information too much that it cannot transform the style enough. In contrast, VQGAN-NADA and CLIPstyler modulate the images too much that they even transform the content. However, our method could achieve both of them. Meanwhile, CLIP score is calculated in a global manner as explained in the equation (7) and in a patch-based manner. Face identity loss is calculated using ArcFace (Deng et al., 2019). The images used for quantitative experiments are the same as those used for the user study.

### 4.2.2 DIFFUSION MODELS

For diffusion-based models, we adopt two methods for comparison - DDIB (Su et al., 2022) and DiffusionCLIP (Kim et al., 2022). Since DDIB requires two different score functions, we trained a new diffusion model on Wikiart dataset in order to evaluate its translation performance between painting and photo domains. For the photo domain, we utilized the pretrained diffusion models described above. The qualitative and quantitative results of the comparison are demonstrated in the Figure 5 and the Table 2.

The third column of the Figure 5 shows that DDIB suffers from identity loss. Though the portrait is translated into photo, its facial identity is destroyed. Also, it is hard to delineate the shape of the church with the output of DDIB. Furthermore, the most critical drawback of DDIB is that diffusion models have to be trained for each new domain. In this regard, image translation from portrait to neon light style is not available. Meanwhile, DiffusionCLIP shows relatively satisfying quality in translating photos into another style. Once the images are not well converted into the photo domain, however, it is inevitable to get unsatisfactory results from unseen domain images, as shown in the first and third rows of the Figure 5. This is supported by user study results on DiffusionCLIP. As can be seen in the Table 2, its content score in unseen domains is 0.85 lower than the score in photo domain. More examples are provided in the Figure 17. In contrast, we can observe that our proposed method can stylize images not only from photo domains but also unseen domains such as portrait or painting. The portrait is transformed into photo while maintaining its facial identity. Also, the painting of church is translated into neon light style while retaining even small objects

like a cross. It is confirmed with user study results. The scores between photo domain and unseen domains are highly similar, which means that our method can modulate images even from unseen domain. In terms of computational time, as shown in Table 5, our method is significantly faster than DiffusionCLIP.

### 4.3 ROLES OF CONTENT LOSSES

In order to check the roles of content guidance losses, we conducted ablation studies. As proposed in the equation (11), the content loss for guidance is comprised of three different losses, $L_{CUT}$, $L_{VGG}$, and $L_{MSE}$. Among the three losses, we investigate the power of $L_{CUT}$ by eliminating it from the total content loss. In Figure 6, we can see that $L_{VGG}$ and $L_{MSE}$ are not enough to sufficiently preserve the structural properties. The outlines of the buildings remain, but not their details such as windows. When we employ all the three losses, we could get the best results regarding content preservation. In the upper row of the Figure 6, even letters in the source image are preserved in the output with contrastive content loss. The superiority of $L_{CUT}$ is also justified with user study results. As shown in the Table 3, $L_{CUT}$ scores higher than $L_{MSE}$, $L_{VGG}$, and even both of them.

### 4.4 SKIPPING MLP

As discussed before, we also removed the MLP network $F$ to eliminate the fine-tuning process. As shown in the Figure 7 and the Table 4, the quality of the generated images is slightly inferior to our method in perspective of both content preservation and style modulation. This justifies the use of MLP network $F$ to extract better features. However, the difference in quality is so small that users can choose which method to use based on their preferences.

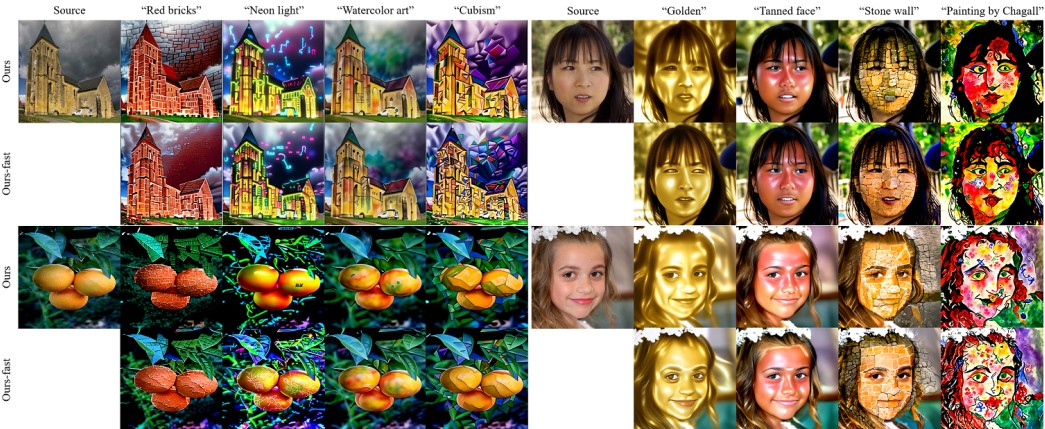

Figure 7: Style-transferred images from the fast version of our method.

| Methods | CLIP score ↑ | VGG loss ↓ | Computation time (sec) |
|---------|--------------|------------|------------------------|
| Ours | **0.1705** | **90.3081** | 91 |
| Ours-fast | 0.1643 | 96.8257 | **45** |

Table 4: Quantitative comparisons between our method and its fast version.

## 5 CONCLUSION

In this paper, we proposed a diffusion based image style transfer without content changes by utilizing the contrastive loss. Our method does not require additional training on the diffusion models, and only requires light MLP training. Furthermore, our fast implementation does not require any MLP layers so that the computational time is significantly reduced with a slight performance loss. Extensive experiments demonstrated that contrastive loss with diffusion model results in the high capability to maintain content. The discussions on limitations are given in the Appendix.

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

## A  DETAILS OF IMPLEMENTATION

### A.1  DATA MANIPULATION AND HYPERPARAMETERS

In order to utilize directional CLIP loss in a patch-based manner (Kwon & Ye, 2022), we randomly crop the source image. The patch size can have various ranges, but we mainly utilize $(0.01, 0.05)$ for texture styles, such as golden or green crystal, and $(0.01, 0.3)$ for artistic styles, such as painting by Gogh or pop art. The cropped images are then augmented with perspective function and random affine transformation.

For style and content guidance, we utilized various weights for each loss according to different styles. Even though the values can be varying for each style, the weights for $L_{global}$ and $L_{directional}$ usually range from 5000 to 30000. In addition, the weights for $L_{CUT}$, $L_{VGG}$, and $L_{MSE}$ are generally 100, 100, and 10000, respectively. However, one can change these values to improve image quality.

### A.2  PATCH-WISE CROSS ENTROPY LOSS FOR CUT GUIDANCE

We explain cross-entropy loss in the equation (9) in more detail (Park et al., 2020). The inputs for the loss are a query $v$, and its positive $v^+$ and negatives $v_i^-$ where $i \in [1, \ldots, N]$ . The query is a patch from the generated image and the positive is the corresponding patch of the source image. The negatives are the other non-corresponding patches of the source image. Then the cross-entropy loss helps a patch share embedding space with the corresponding patch of the input and not with the other patches. It can be written as,

$$\ell(v, v^+, v^-) = -log\left[\frac{e^{v \cdot v^+/\tau}}{e^{v \cdot v^+/\tau} + \Sigma_{i=1}^N e^{v \cdot v_i^-/\tau}}\right] \tag{13}$$

where $\tau$ is a temperature

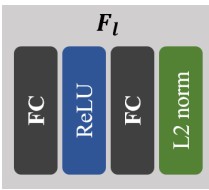

Figure 8: Diagram of MLP network $F_l$. Its input is feature maps from the $l$-th layer of the encoder part of the noise estimator $\epsilon_\theta$.

### A.3  TRAINING DDIB

For comparison with DDIB, we trained a diffusion model with 13 thousands of images from Wikiart dataset. They were $256 \times 256$ in size. The architecture of the model is based on the guided diffusion (Dhariwal & Nichol, 2021). The model use 128 base channels and the attention at $16 \times 16$ and $8 \times 8$ resolutions. Residual blocks for upsampling and downsampling are not used. We fixed the variance as a constant (Ho et al., 2020). The model was trained during 50,000 iterations with batch size $8$ on a NVIDIA RTX 3090.

## B  ADDITIONAL EXPERIMENTAL RESULTS

### B.1  EFFECT OF THE NUMBER OF TIMESTEPS

Since the diffusion process usually takes lots of time, two techniques are widely used - respacing and skipping time steps (Chung et al., 2022; Kim et al., 2022). The last time step $T$ is respaced into $T'$. Then we forward the diffusion model to time $t_0 < T'$ and reverse the diffusion process from $x_{t_0}$. $T'$ and $t_0$ have various effects on both image quality and time consumption. As shown in the

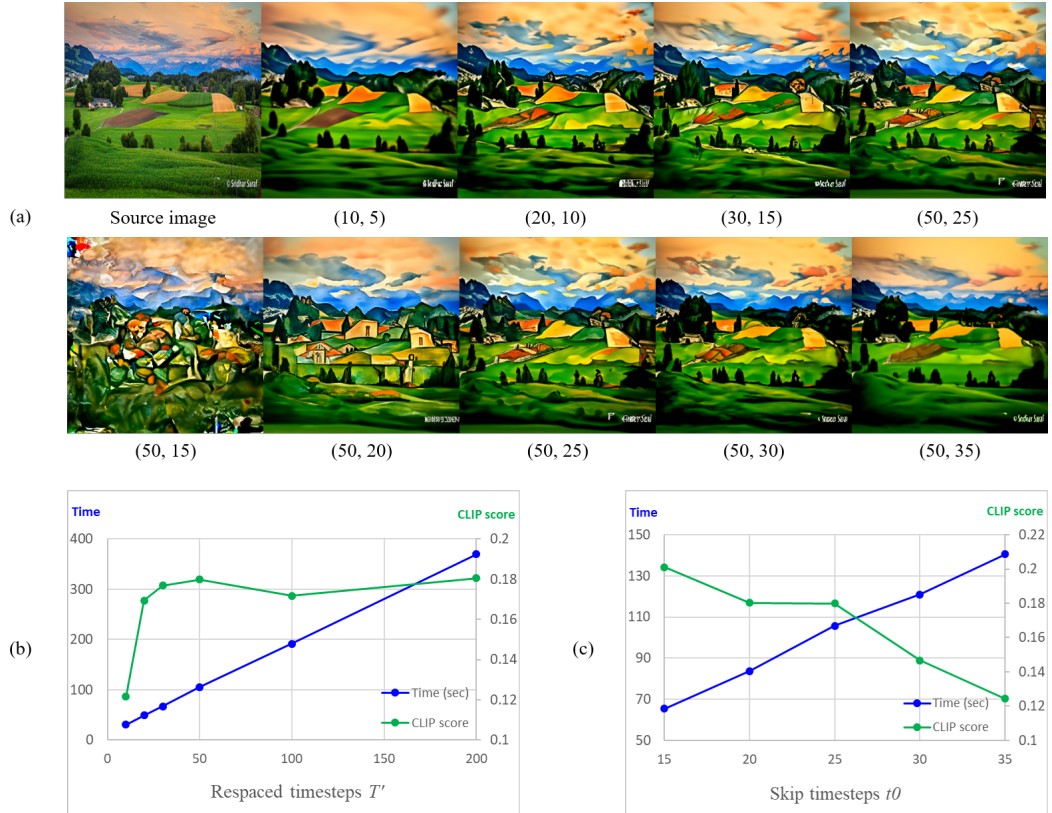

Figure 9: The effect of the respaced time $T'$ and skipped time $t_0$. (a) demonstrates the images sampled with $(T',t_0)$. The first row shows the difference between various $T'$ when $t_0$ is its half and the second row shows the difference between various $t_0$ when $T' = 50$. (b) and (c) illustrates the relationship between sampling time and CLIP score as graphs for the first and second rows of (a), respectively.

Figure 9 (a) and (b), image quality with respect to style transformation enhances as respacing time step $T'$ increases. However, its growth rate decreases and its difference is imperceptible even though sampling time still increases. In the mean time, CLIP score decreases as skip time step $t_0$ increases as illustrated in the Figure 9 (c). On the other hand, the content information is not fully preserved in the early time steps $t_0 = 15$ or 20 as shown in the Figure 9 (a). Thus, we set $(T', t_0)$ as (50, 25) for our baseline.

## B.2 Diffusion models' trade-off between style and content

With respect to style transfer, one of the challenges posed by unconditional diffusion models is to maintain content of the given image. When transforming styles of the given image, its content changes simultaneously. GAN-based methods explicitly impose content losses, such as a reconstruction loss. This results in good performance in content preservation. In contrast, diffusion models have no constraint during training phase. They generate high quality images in correspondence with the training data domain. The semantic constraints are not considered which finally results in the degradation in the quality of the generated images.

Here, we compare four diffusion models - ILVR, DDIM, DiffusionCLIP, and our proposed method - with respect to style and content in the Figure 2. ILVR utilizes down-sampled reference image as condition in each reverse denoising steps. The condition helps the generated image share its content information with the reference image. However, it cannot have same identity because reverse DDPM steps without condition should be given sufficiently in order to generate images in photo style. This accordingly results in a loss of content. DDIM can reconstruct the source image when the variance

| Methods | Data preparation | # Param. | Training time | Inference time (sec) |
|---|---|---|---|---|
| ILVR | - | - | - | 100 |
| DDIM | - | - | - | 11 |
| DDIB | - | 1104 M | > 200 hrs | 12 |
| DiffusionCLIP | 5.85 min | 113 M | 293 sec | 96 |
| Ours | - | 0.7 M | 45 sec | 45 |
| Ours-fast | - | - | - | 45 |

Table 5: Comparison on computational complexity of the diffusion models.

of noise $\sigma_t$ is set as $0$. However, the style is also preserved with zero variance. When we control $\sigma_t$ as larger than zero, we can get photo style images but their content is altered with stochastic noise. DiffusionCLIP tried to solve the trade-off between content and style by fine-tuning the score function $\epsilon_\theta$. However, it requires much more time due to the model training for each style and data preparation. In addition, the content cannot be maintained when the source images are from unseen domains. In contrast, our proposed method does not require additional training on the diffusion model. This leads to shorter time than DiffusionCLIP. With the help of CUT guidance, we could retain the content of source image from any domain and translate it into different styles.

## B.3 MORE COMPARISON WITH GAN AND CNN-BASED METHODS

In addition to the comparative studies on the GAN-based methods in the Section 4.2.1, we conducted more comparisons with various GAN-based and CNN-based methods including both text and image guidance. For text guidance method, we compared our method with one more method, LDAST (Fu et al., 2022). For image guidance, we included three methods, SANet (Fan & Ling, 2017), AdaIN (Huang & Belongie, 2017), and WCT2 (Yoo et al., 2019). As shown in the Figure 10, we could notice that LDAST and WCT2 could preserve the content information better than SANet and AdaIN. However, all the four methods for comparison show inferior performance in perspective of style transformation.

## B.4 ROLES OF LOSSES

Our loss consists of $L_{global}$, and $L_{directional}$ as shown in the equation (6). We examine the role of each loss by applying the loss functions one by one. In Figure 11, we evaluate on three styles - green crystal, neon light, and fire. The images in the lower row tend to be more stylized than images in the upper row, which means that directional CLIP loss leads the diffusion models to higher performance in style modulation. In addition, we verified the role of patch-based guidance. The results show

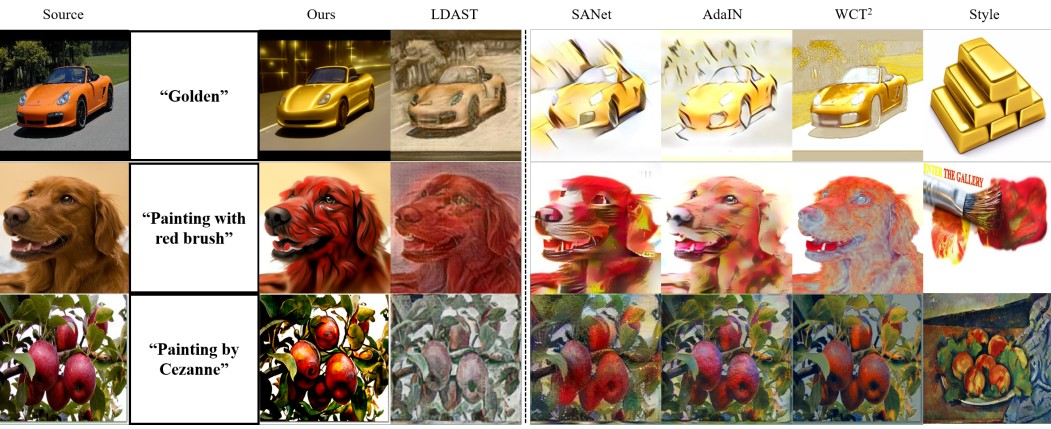

Figure 10: Comparative study results.

that CLIP guidance with whole image tends to stylize the image in local parts. In contrast, the patch-based guidance transforms the image into the given style that covers large area.

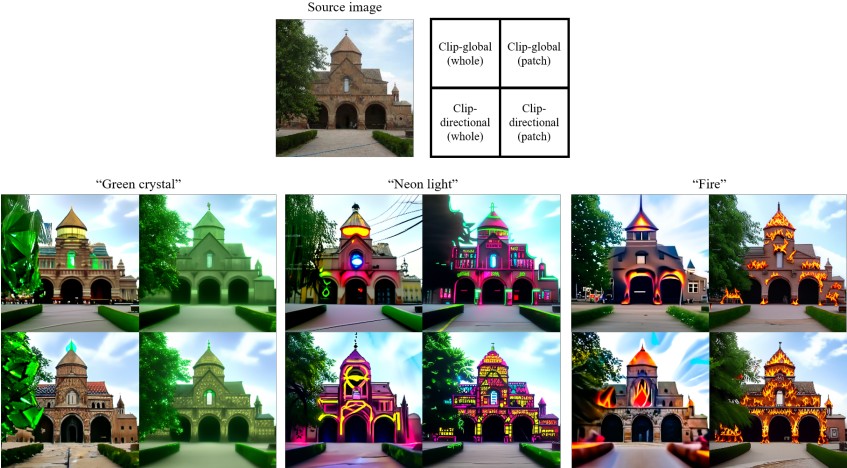

Figure 11: Ablation study on four losses for style guidance - CLIP global loss, patch-based CLIP global loss, CLIP directional loss, and patch-based CLIP directional loss.

## B.5 DDPM AND DDIM FOR DIFFUSION PROCESSES

Although either DDPM or DDIM can be utilized for both forward and reverse processes, we conducted a comparative study in order to show their differences in the generated images. As shown in Figure 12, results from the forward DDIM show better performance in preserving content than DDPM. For reverse process, DDPM tends to transform styles better compared to DDIM. Accordingly, we chose to use DDIM as forward and DDPM as reverse process as default.

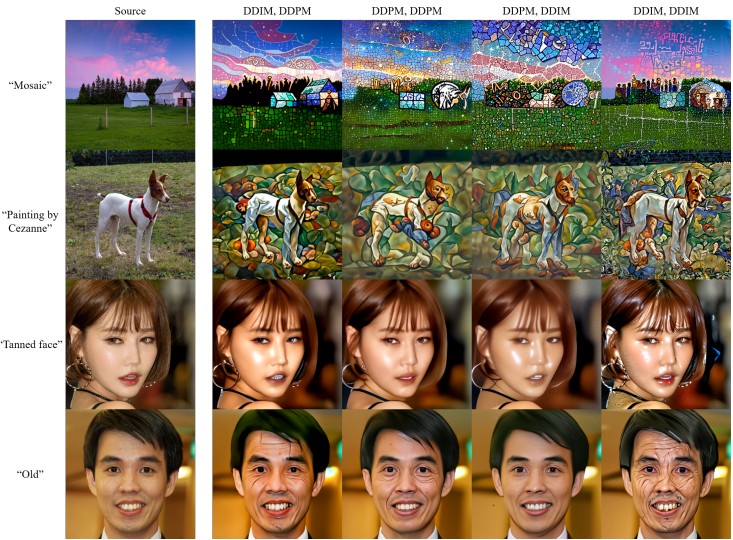

Figure 12: Ablation study results on diffusion processes. From the second column to the right, the combinations of methods (forward, reverse) are (DDIM, DDPM), (DDPM, DDPM), (DDPM, DDIM), and (DDIM, DDIM).

### B.6 UNSEEN DOMAINS

DiffusionCLIP tried to solve the trade-off between content and style by fine-tuning the diffusion model with identity loss. Because of the constraints imposed on the finetuned model, the transformed image shows high performance in identity preservation. However, the fine-tuned model $\hat{\epsilon}_\theta$ converts only the photo domain images. When it comes to unseen domains, such as portraits or paintings, they should be converted to photo images through $\epsilon_\theta$. Since it has not been fine-tuned with identity loss, the semantic information is lost during the reverse sampling process due to the stochastic property of the diffusion model. Thus, the final output from images of unseen domains is degraded in its quality. In contrast, our proposed method can transform even the unseen domain images with only one step. Since our method can preserve the identity with content guidance, the final outputs do not suffer from quality degradation. Also, our method takes about 90 seconds whereas DiffusionCLIP requires about 400 seconds for model fine-tuning and sampling. As described in the Figure 2, DiffusionCLIP requires two steps from portrait to photo to Pixar domains. In this process, the face identity is destroyed. However, the proposed method could preserve the identity while transforming into the style of Pixar.

### B.7 USER STUDY

For quantitative analysis, we conducted user study. For comparison with GAN-based methods, 60 images with four styles have been used in total. The styles involved are "golden", "clay", "3d render in the style of Pixar", and "pop art." We utilized human face images because StyleCLIP and StyleGAN-NADA are based on face dataset. In addition, we totally generated 24 images with six styles for comparison with DiffusionCLIP. We chose three styles ("neon light", "green crystal", and "Ukiyo-e") for the photo domain and the other three styles ("3d render in the style of Pixar", "pop art", and "golden") for unseen domains. We used portraits and paintings from Wikiart dataset for unseen domain images. Besides, for ablation study on content losses, we used 15 images for three styles ("golden", "oil painting of flowers", "leather") in total. The number of questions were 14. 20 users participated in the user study, and their ages range from 20 to 60 years old. They were randomly recruited online.

## C LIMITATIONS

Although our proposed method has various strengths and shows great performance, there remain some limitations. As described in the Appendix A.1, one should find weights for each loss though their relevant ranges are given in this paper.

| Model | Style prompt | CLIP-global | CLIP-directional | CUT | MSE | VGG | Patch size | $t_0$ |
|---|---|---|---|---|---|---|---|---|
| ImageNET | Cubism | 30000 | 40000 | 1000 | 10000 | 50 | 0.05 | 25 |
| | Watercolor art | 10000 | 10000 | 200 | 0 | 100 | 0.05 | 25 |
| | Ukiyo-e | 10000 | 20000 | 200 | 30000 | 200 | 0.3 | 30 |
| | Oil painting of flowers | 20000 | 20000 | 1200 | 10000 | 10 | 0.05 | 25 |
| | Red bricks | 20000 | 50000 | 1000 | 1000 | 10 | 0.05 | 25 |
| | Wooden | 20000 | 30000 | 500 | 10000 | 100 | 0.05 | 25 |
| | Leather | 20000 | 50000 | 1000 | 1000 | 10 | 0.05 | 25 |
| | Marbling | 20000 | 20000 | 2000 | 20000 | 200 | 0.3 | 25 |
| | Autumn | 20000 | 20000 | 700 | 10000 | 100 | 0.05 | 25 |
| | Snowy | 20000 | 20000 | 700 | 10000 | 100 | 0.05 | 25 |
| FFHQ | 3d render in the style of Pixar | 7000 | 7000 | 500 | 10000 | 100 | 0.3 | 25 |
| | Pop art | 20000 | 30000 | 100 | 300 | 70 | 0.3 | 25 |
| | Ukiyo-e | 20000 | 30000 | 800 | 3000 | 50 | 0.3 | 25 |
| | Stone wall | 20000 | 40000 | 2000 | 10000 | 10 | 0.1 | 25 |
| | Red bricks | 20000 | 20000 | 500 | 10000 | 100 | 0.05 | 25 |
| | Wooden | 20000 | 20000 | 500 | 10000 | 100 | 0.05 | 25 |
| | Leather | 20000 | 30000 | 500 | 10000 | 10 | 0.05 | 25 |
| | Clay | 40000 | 40000 | 1000 | 10000 | 0 | 0.05 | 25 |
| | Stained glasses | 20000 | 40000 | 1000 | 10000 | 10 | 0.3 | 25 |
| | Golden | 10000 | 10000 | 500 | 0 | 100 | 0.05 | 15 |

Table 6: Examples of hyperparameters for various style prompts. Weights for CLIP-global loss, CLIP-directional loss, CUT loss, MSE loss, and VGG loss are given. For patch-based CLIP guidance, we control the patch size. The maximum size is given in the table with the minimum of 0.01. $t_0$ is the time step to which the source image is forwarded when $T' = 50$.

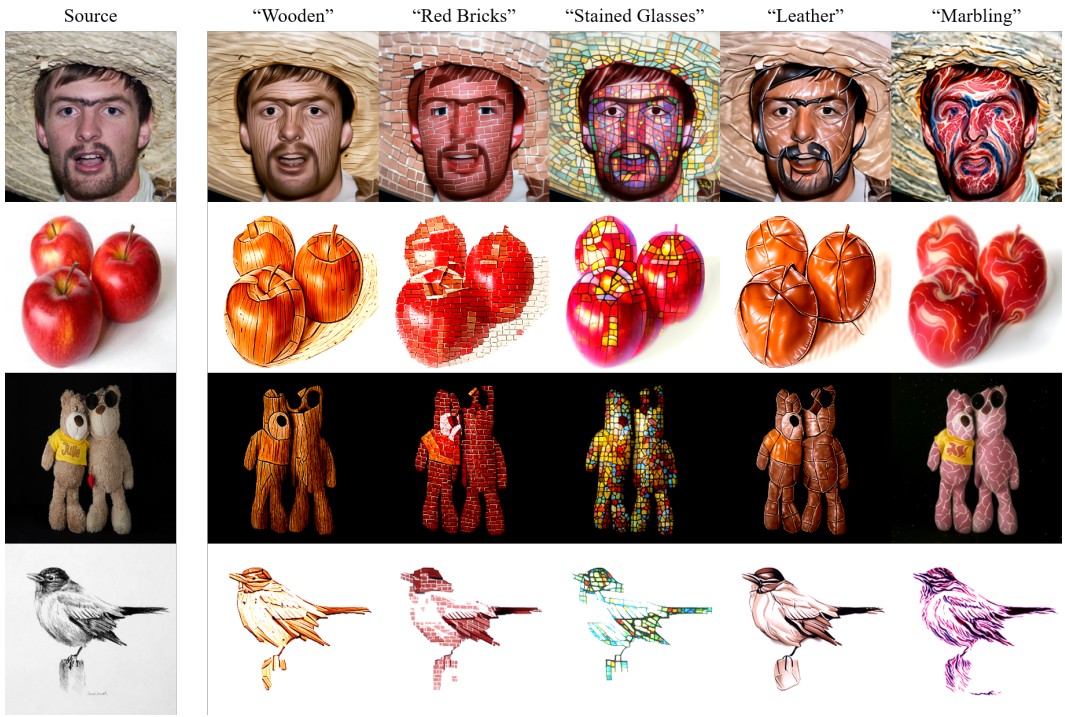

Figure 13: Additional results on various style prompts.

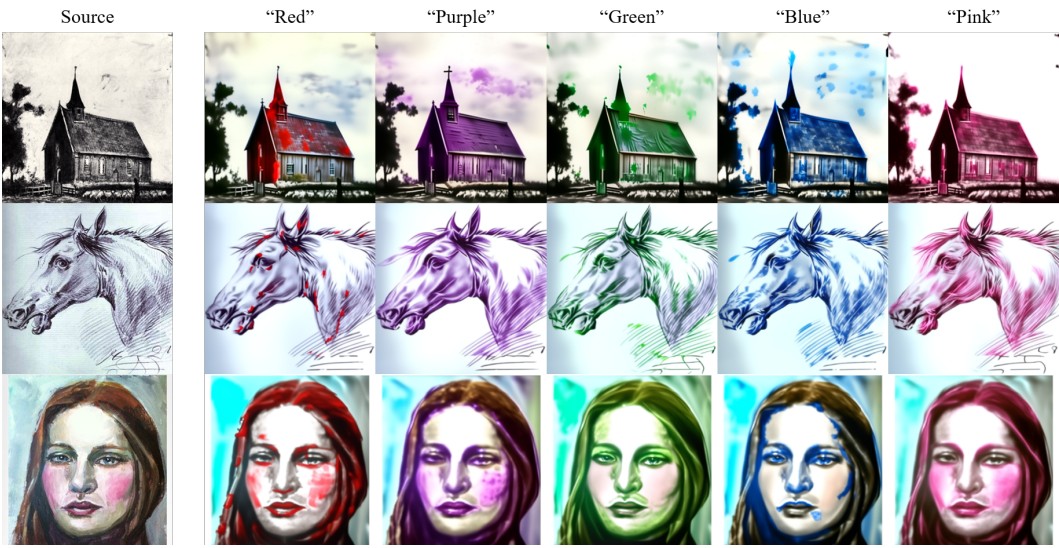

Figure 14: Additional results on color style prompts.

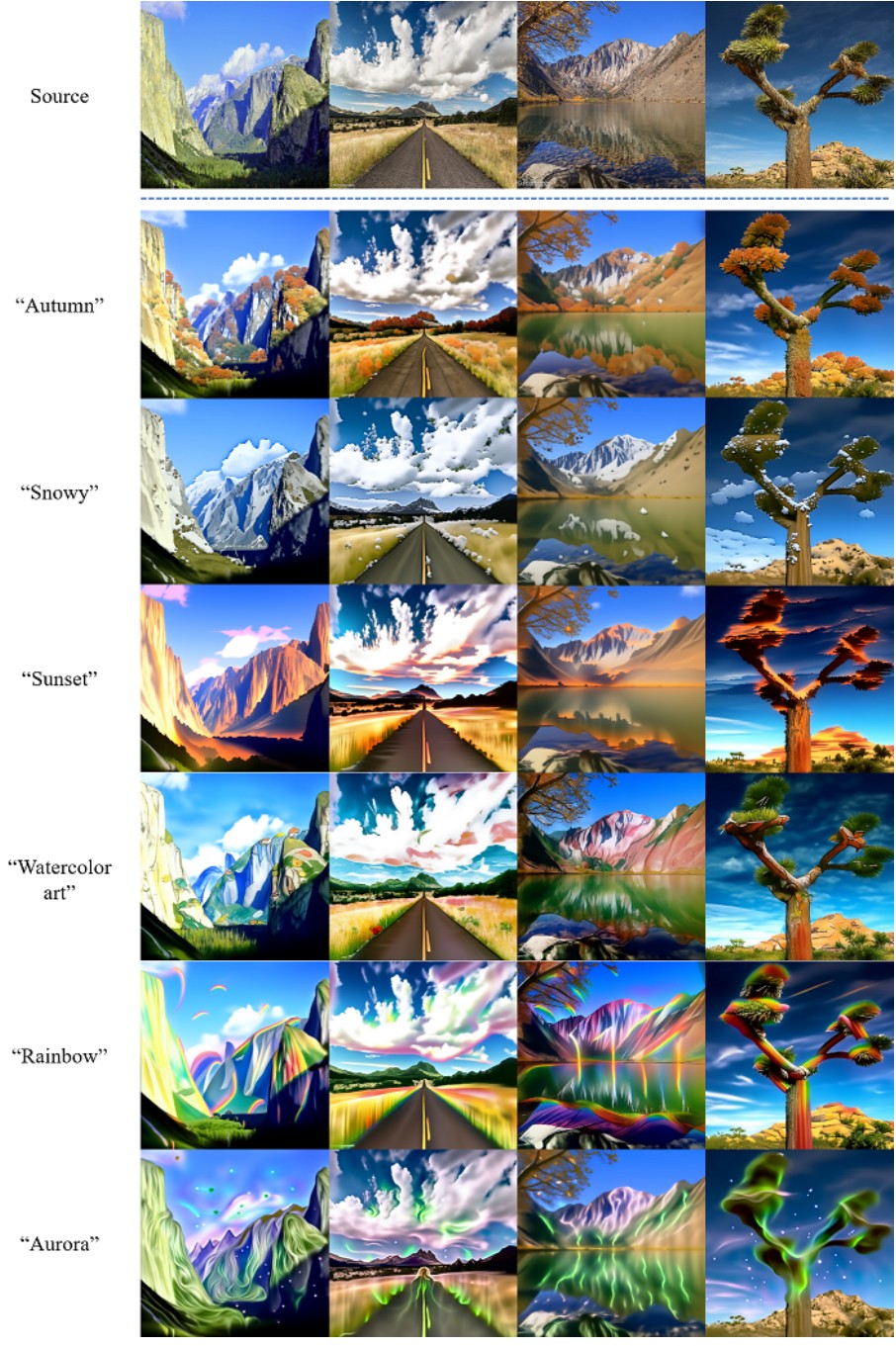

Figure 15: Additional results on various style prompts.

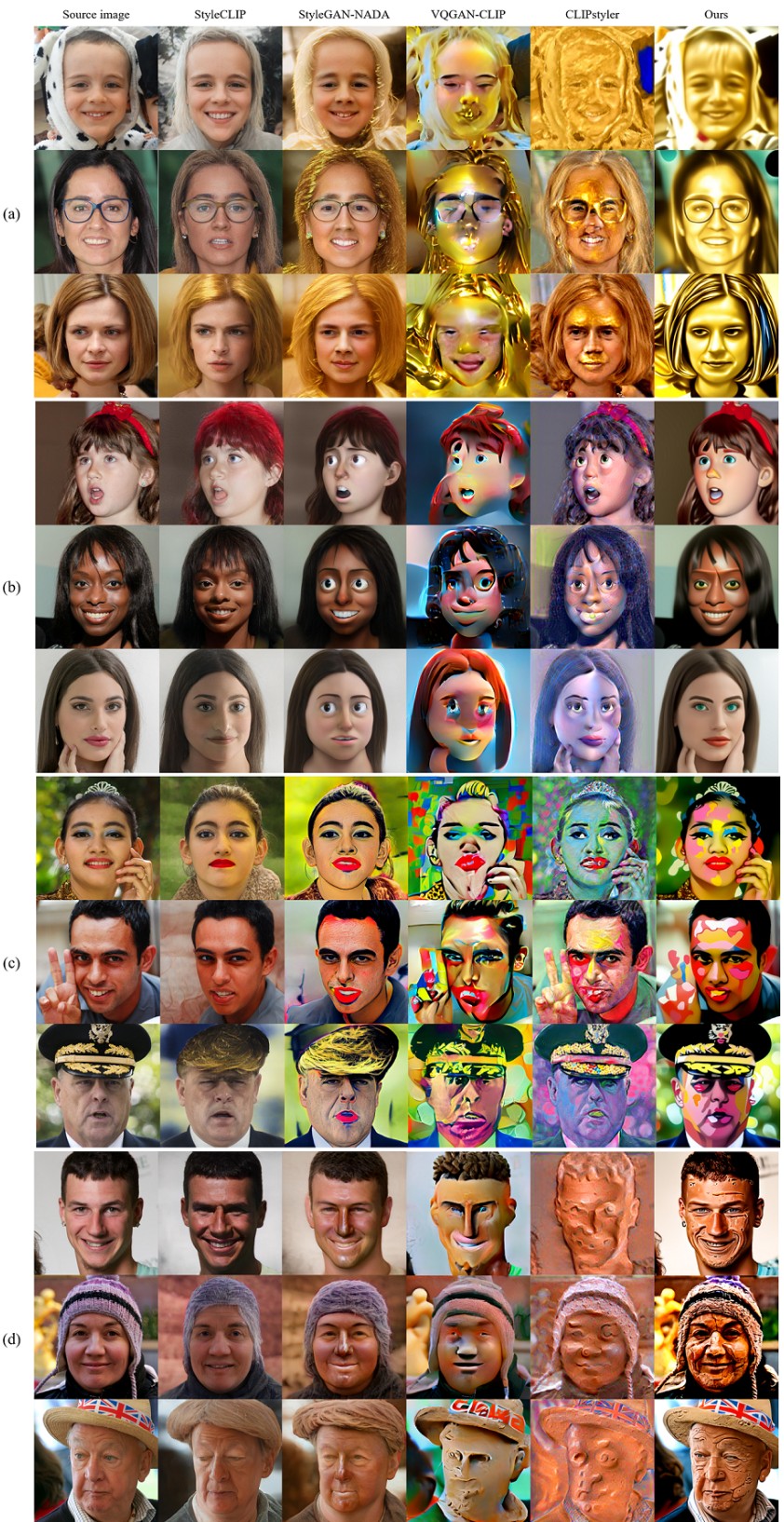

Figure 16: Additional results on the comparative studies. (a), (b), (c), and (d) are the results on styles of "golden", "3d render in the style of Pixar", "pop art", and "clay", respectively.

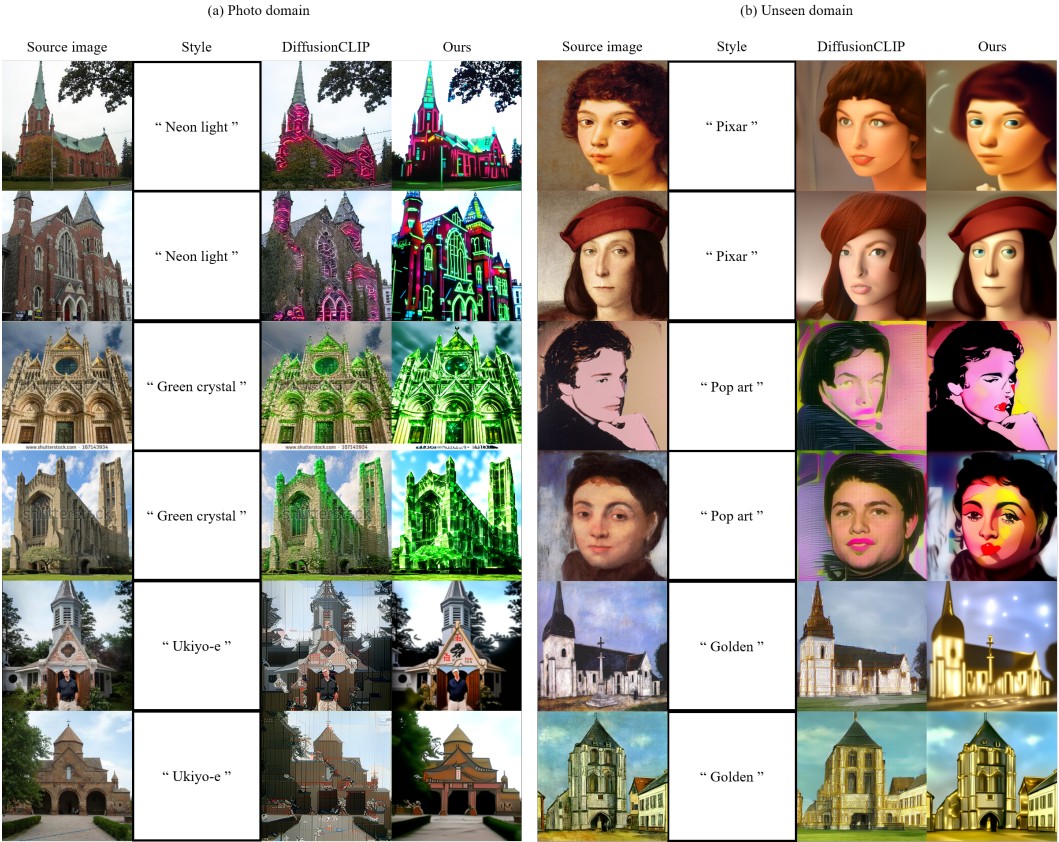

Figure 17: Additional results on the comparative studies. Source images in (a) are from photo domain and ones in (b) are from unseen domains such as portrait or painting.

