# OpenReview forum: "Text-Guided Diffusion Image Style Transfer with Contrastive Loss Fine-tuning"
_ICLR.cc/2023/Conference — Submitted to ICLR 2023_

### Official Review · Reviewer_n6KZ · 2022-10-21

**Confidence:** 3
**Correctness:** 3
**Technical Novelty And Significance:** 2
**Empirical Novelty And Significance:** 2
**Recommendation:** 5

**Clarity, Quality, Novelty And Reproducibility:**

This paper is well-written and can be easily reproduced.
The reviewer found this paper was accepted by ECCV 2022 Workshop on Uncertainty Quantification for Computer Vision. The authors should explain this issue seriously.

**Strength And Weaknesses:**

Strength

+Maintaining content well
In contrast to the previous methods, the proposed can preserve the content well. However, it is not clear if the proposed method outperforms previous CNNs- or GANs-based style transfer methods. These methods commonly preserve content well. It also means that more methods should be included for comparison, not just comparing with StyleCLIP, StyleGAN-NADA, and VQGAN-CLIP.

+Insight on style transfer with the unconditional diffusion model.
This work presents some insights into how to implement style transfer with the unconditional diffusion model. Moreover, it does not require retraining the diffusion model.

+Well-written
This paper is well-written and easy to understand. Especially, Figure 2 clearly presents the motivation and difference of the proposed method.


Weaknesses

-Limited style
In spite of using the CLIP model, it seems the proposed method can only obtain limited text styles such as Pixar, Pop art, Ukiyo-e, etc. All the testing results use these texts. How about other styles? More diverse results should be included to show the effectiveness of the proposed method.

-Limited style transfer quality.
As shown in the main paper and appendix, the quality OF transferred styles is not much promising. The reviewer knows that the proposed outperforms other compared methods in the figures shown. However, the quality is still not satisfactory.

-Limited novelty.
As the core of this work, the conservative loss with diffusion model is similar to CUT which is used in condition image synthesis. The differences or modifications are not clear.

-Insufficient analysis and experiments
Besides the limited texts and examples, the computational complexity comparison is missing. This work highlights that the computational complexity is much low. Some analysis and quantitative results are expected.

**Summary Of The Paper:**

This paper proposes a text-guided diffusion model for style transfer. To achieve that, a text-guided sampling scheme i.e., patch-wise contrastive loss fine-tuning was used. Compared with previous related works, the proposed method can preserve the semantic content of the source image well. Besides, the proposed method requires no additional training on the diffusion model.

**Summary Of The Review:**

The paper has some novelty and experiment flaws, which are the most important factors in my rating. The authors can provide comments based on the points I mentioned in the weaknesses. After that, I will consider my final decision.

---

> ### Author Response · Authors · 2022-11-16
> **Response to Reviewer n6KZ (2/2)**
>
> > **W5: Besides the limited texts and examples, the computational complexity comparison is missing. This work highlights that the computational complexity is much low. Some analysis and quantitative results are expected.**
> * We appreciate your constructive comments. Due to the iterative nature,  it is difficult to evaluate computational complexity of diffusion models in terms of FLOPs or MACs. Instead, we include comparative studies on training and inference time and the number of parameters.
> * Please refer to the Table 5. Diffusion model is notorious for its long training and inference time. DDIB needs to train two independent score functions which make its computational cost expensive. DiffusionCLIP therefore tried to lessen the computation complexity by finetuning the pretrained score function. However, our method does not require any training on the diffusion model. Instead, we only fine-tune the MLP layers which are much lighter. Moreover, this MLP fine-tuing is not even necessary in our fast version. Therefore, our method shows much shorter training time and smaller number of parameters.
>
> > **W6: The reviewer found this paper was accepted by ECCV 2022 Workshop on Uncertainty Quantification for Computer Vision. The authors should explain this issue seriously.**
> * We suspect that the reviewer confused our paper with others. To our best knowledge, this is the paper’s first submission and we have not found any similar papers from others in ECCV 2022 Workshop.

---

> ### Author Response · Authors · 2022-11-16
> **Response to Reviewer n6KZ (1/2)**
>
> > **W1: In contrast to the previous methods, the proposed can preserve the content well. However, it is not clear if the proposed method outperforms previous CNNs- or GANs-based style transfer methods. These methods commonly preserve content well. It also means that more methods should be included for comparison, not just comparing with StyleCLIP, StyleGAN-NADA, and VQGAN-CLIP.**
> * Thanks for your positive comments. We have already shown comparisons with GAN-based methods such as StyleCLIP, StyleGAN-NADA, VQGAN-CLIP and CLIPstyler, which confirm the advantages of our diffusion models.  In order to more support the strength of our proposed method, we included more CNNs- or GANs- based style transfer methods for comparison; LDAST, SANet, AdaIN, and WCT2.
> * As shown in the Figure 4 and 10, VQGAN-CLIP, CLIPstyler and LDAST show good performance at preserving content. However, it can be noticed that the eyes are crushed from human faces. Moreover, the styles are not modulated enough.  As illustrated in the Figure 10, we also included image guidance methods such as SANet, AdaIN, and WCT2. However, compared with our proposed method, they show inferior results in both style transformation and content preservation.
>
> > **W2: In spite of using the CLIP model, it seems the proposed method can only obtain limited text styles such as Pixar, Pop art, Ukiyo-e, etc. All the testing results use these texts. How about other styles? More diverse results should be included to show the effectiveness of the proposed method.**
> * Thanks for your valuable comments. In the paper, we have already applied 17 style prompts including artistic and texture-related styles. In addition to these prompts, we have now added results from 20 more style prompts including weather (autumn, snowy, sunset), face (tanned face, old), color (red, blue, green, purple, pink), and other texture- related and artistic prompts (wooden, red bricks, stained glasses, stone wall, rainbow, aurora, painting by Chagall, marbling, leather, etc.). Please refer to the Figure 12, 13, 14, 15.
>
> > **W3: As shown in the main paper and appendix, the quality OF transferred styles is not much promising. The reviewer knows that the proposed outperforms other compared methods in the figures shown. However, the quality is still not satisfactory.**
> * Our proposed method has shown promising results on not only quantitative results but also human evaluation. In addition, from extensive experiments, we show that our method consistently works with a variety of style prompts.
>     * When comparing CLIP score and Face identity loss, our method shows better results compared to GAN-based methods. (Figure 4, 10, and Table 1).
>     * We have shown that our method even works with images from unseen domain such as painting. (Figure 5 and 17)
>     * The quality has also been proved by the human evaluation. Our method scored more than 4.0 out of 5.0 in both style transformation and content preservation. (Table 1 and 2)
>     * This good performance has been proved with a wide variety of style prompts. The generated images show consistently decent quality from the Figure 13, 14, and 15.
>
> > **W4: As the core of this work, the conservative loss with diffusion model is similar to CUT which is used in condition image synthesis. The differences or modifications are not clear.**
> * Despite the diffusion model’s great performance in generating diverse images, the stochastic nature of reverse diffusion hinders maintaining content, which has been a limitation for style transfer task.  The main motivation of our paper is to exploit the spatial features from diffusion models using contrastive loss so that we can maintain the semantic content during the style transfer. Although similar idea of using contrast loss has been used in conventional image generation, to our best knowledge, this is the first work that applies the similar observation to diffusion models, whose contribution should not be disregarded.
> * Furthermore, as discussed in the General Comments, we have shown that even the fine-tuning with MLP is not necessary with a slight performance loss,  as the diffusion model already have spatial features that can be used for contrast loss to maintain the semantic content.

---

### Official Review · Reviewer_VPPF · 2022-10-26

**Confidence:** 3
**Correctness:** 4
**Technical Novelty And Significance:** 2
**Empirical Novelty And Significance:** 3
**Recommendation:** 5

**Clarity, Quality, Novelty And Reproducibility:**

Some parts are a bit difficult to understand:
1. I feel difficult to understand the text describing how the MLP is fine-tuned below Eq.10.
2. "The DDIM strategy as the forward noising process and DDPM method as the reverse sampling", what's the benefit of doing so?
3. Network $F_l$ (before Eq.10) could be illustrated with a diagram.
4. In the bottom of Page 6, "with T' = 50, we forward the source image x0 to latent image x_t0 where t0 = 25", what does this mean?



**Strength And Weaknesses:**

Strengths:
The empirical results seems not bad. The generated images faithfully keep the contents of the original images, and the styles are consistent with the text prompt.

Weaknesses:
1. The method proposed in this paper is basically a combination of a few losses proposed before, i.e., the CLIP global loss and directional loss proposed in StyleGAN-NADA, and the CUT loss proposed in (Park, ECCV 2020). I feel that the novelty is rather limited.
2. Although the paper is easy to follow overall, some parts are a bit difficult to understand, see the section below.
3. Some grammar/format errors:
 * Parenthetical citations and citations within the text are incorrectly used in Second paragraph, Introduction.
 * (Missing "The") CLIP model is trained... (text before Eq.6)
 * the global loss has shortcomings of mode collapse => the global loss suffers from mode collapse


**Summary Of The Paper:**

This paper proposes to do text-guided style transfer using diffusion models with a combination of a few losses, including two CLIP losses and Contrastive Unpaired Translation (CUT) loss. The CLIP global loss and directional loss guide the image generation towards the desired style specified by the prompt, and the CUT loss encourages the generated image to retain the content of the source image. The results look not bad. But the novelty seems to be limited.

**Summary Of The Review:**

The paper investigates a synergetic combination of a few losses that both guide with the desired new style and enforce the existing contents.  It's a useful work, but the novelty is quite limited, as it's a straightforward combination of existing losses, and there's little insight shed onto the generation process.

---

> ### Author Response · Authors · 2022-11-16
> **Response to Reviewer VPPF**
>
> > **W1: The method proposed in this paper is basically a combination of a few losses proposed before, i.e., the CLIP global loss and directional loss proposed in StyleGAN-NADA, and the CUT loss proposed in (Park, ECCV 2020). I feel that the novelty is rather limited.**
> * We appreciate your valuable comments. Regarding the novelty and difference from the previous approaches and motivations, please refer to the general comments above. Although the diffusion model’s great performance in generating diverse images, the stochastic nature of reverse diffusion hinders maintaining content, which has been a limitation for style transfer task.  The main motivation of our paper is to exploit the spatial features from diffusion models using contrastive loss so that we can maintain the semantic content during the style transfer. Although similar idea of using contrast loss has been used in conventional image generation, to our best knowledge, this is the first work that applies the similar observation to diffusion models, whose contribution should not be disregarded.
>
> > **W2: Some grammar/format errors**
> * Fixed.
>
> > **W3: I feel difficult to understand the text describing how the MLP is fine-tuned below Eq.10.**
> * Thanks for your constructive comments. To avoid the confusion from illustration in the Section 3 (Method), we have made significant improvements with detailed explanation. Furthermore, as discussed in the General Comments, we have shown that even the fine-tuning with MLP is not necessary with a slight performance loss because the diffusion model already has spatial features that can be used for contrast loss to maintain the semantic content.
>
> > **W4: "The DDIM strategy as the forward noising process and DDPM method as the reverse sampling", what's the benefit of doing so?**
> * As explained in the Section 4.1, either DDPM or DDIM can be applied for noising and denoising processes. However, in order to show the differences between four combinations, we conducted a comparative study. Please refer to the Figure 12. In perspective of the forward noising process, DDIM is better at preserving content than DDPM. For the reverse sampling process, DDPM shows more decent quality compared to DDIM, which is why we explore the best of the two worlds.
>
> > **W5: Network $F_l$ (before Eq.10) could be illustrated with a diagram.**
> * $F_l$ is the network consisting of two MLP layers, ReLU activation function, and L2 normalization layer. For easier understanding, we added the diagram of $F_l$ in the Figure 8. The detailed explanation is given in the Method Section.
>
> > **W6: In the bottom of Page 6, "with T' = 50, we forward the source image x0 to latent image x_t0 where t0 = 25", what does this mean?**
> * We appreciate your valuable comment. We elaborated our explanation in the paper for clear understanding.
> * $T’$ is the number of respaced time steps and $t_0$ is the intermediate time step to which the source is forwarded. We first respace the total time steps 1000 ($T$) into 50 ($T’$). Then, the source image $x_0$ is forwarded to time step 25 ($t_0$). Then, noisy image $x_{25}$ can be obtained. For more detailed explanation, please refer to the Section 4.1 and the Section B.1

---

### Official Review · Reviewer_JYQf · 2022-10-26

**Confidence:** 4
**Correctness:** 3
**Technical Novelty And Significance:** 2
**Empirical Novelty And Significance:** 3
**Recommendation:** 5

**Clarity, Quality, Novelty And Reproducibility:**

I don't think the current version is clear enough in presentation, and I'm still confused by the details of its approach.

**Strength And Weaknesses:**

Strength:
The results are very good.

Weaknesses:
I think the writing and the degree of novelty are questioned, at least from the current version of the paper. But I suspect that the shortcoming of novelty may also be due to writing.

Writing:
1. I am very confused about the fine-tuning method. I only see this word in the abstract and introduction. And I find nothing about it in the method section (please correct me if I'm wrong). This shows that in the writing of the method, the author still needs to make a lot of efforts to improve. I understand that telling about the diffusion model in such a limited space is difficult, but it shouldn't confuse readers like me very much.
2. For Eq (11), are these three loss functions weighted? What are the weights?
3. Many typos. And many references are wrong. lease check them one-by-one.

Novelty:
According to the paper, the losses are mostly borrowed from previous works. That is fine, but I don't see
enough insights in the paper, e.g., a good understanding of the problem and interpretation or ablation studies. I think this problem is partly caused by the writing. It might be better if the authors could explain more clearly how their approach differs from previous approaches and what motivates them.

**Summary Of The Paper:**

This paper proposes a new text-guided Diffusion-based style transfer method. The proposed method uses the CLIP loss for style guidance and the CUT loss for content guidance. The proposed method also uses the fine-tuning strategy according to the abstract and introduction (but I didn't find any words about it in the method section). The overall results seem good. The CUT loss and CLIP loss work well.

**Summary Of The Review:**

Good results are as Plus. But the writing is not ready for acceptance.

---

> ### Author Response · Authors · 2022-11-16
> **Response to Reviewer JYQf**
>
> > **W1: I am very confused about the fine-tuning method. I only see this word in the abstract and introduction. And I find nothing about it in the method section (please correct me if I'm wrong). This shows that in the writing of the method, the author still needs to make a lot of efforts to improve. I understand that telling about the diffusion model in such a limited space is difficult, but it shouldn't confuse readers like me very much.**
> * Thanks for your constructive comments. To avoid the confusion from illustration in the Section 3 (Method), we have made significant improvements with detailed explanation. Furthermore, as discussed in General Comment, we have shown that even the fine-tuning with MLP is not necessary with a slight performance loss because the diffusion model already has spatial features that can be used for contrastive loss to maintain the semantic content.
>
> > **W2: For Eq (11), are these three loss functions weighted? What are the weights?**
> * The loss functions are weighted. These weights are hyperparameters which depend on the style prompts. It is explained in the Section A.1. However, for easier understanding and reproducibility, we added Table 6 which illustrates hyperparameter examples for each style prompt.
>
> > **W3: Many typos. And many references are wrong. lease check them one-by-one.**
> * Done
>
> > **W4: Novelty: According to the paper, the losses are mostly borrowed from previous works. That is fine, but I don't see enough insights in the paper, e.g., a good understanding of the problem and interpretation or ablation studies. I think this problem is partly caused by the writing. It might be better if the authors could explain more clearly how their approach differs from previous approaches and what motivates them.**
> * We appreciate your valuable comments. Regarding the novelty and difference from the previous approaches and motivations, please refer to the General Comments above. Despite the diffusion model’s great performance in generating diverse images, the stochastic nature of reverse diffusion hinders maintaining content, which has been a limitation for style transfer task.  The main motivation of our paper is to exploit the spatial features from diffusion models using contrastive loss so that we can maintain the semantic content during the style transfer. Although similar idea of using contrast loss has been used in conventional image generation, to our best knowledge, this is the first work that applies the similar observation to diffusion models, whose contribution should not be disregarded.

---

### Author Response · Authors · 2022-11-16
**General Comment by Paper4178 Authors**

Thank you for all the constructive comments from the reviewers. In order to address the comments, we have made the following changes.

1. In order to highlight the flexibility of our method, we additionally introduce a fast implementation of our proposed method that does not require MLP fine-tuning. Although we have already made a great advance in perspective of training time, it still needs more than 30 seconds for fine-tuning the MLP network. However, at the expense of a slight quality reduction, we could remove this fine-tuning process by eliminating the MLP network. Thus, users can choose appropriate method according to their preferences. Please refer to the Section 4.4.


2. Emphasizing the novelty
    * Although our proposed method utilizes the existing loss functions, our novelty can be found from the way the CUT loss is incorporated into the diffusion model. The CUT utilizes a patch-wise contrastive loss while training an image generator. This helped training generator to maintain content information while transforming styles. In contrast, we do not train a generator. Instead, we extract feature maps from the noise estimator $\epsilon_{\theta}$ which has already been trained and fixed. By doing so, we could still get benefits from the spatial information captured by the noise estimator. More detailed explanation is given in the Section 3.2.
    * Because of the stochastic property of the diffusion model, the model is hard to be applied to the style transfer task while preserving the semantic content. However, we first propose to apply contrastive loss into the diffusion model so that it can overcome this content issue and accomplish the style transfer task with the diffusion model.

3. We have improved the illustration about how to fine-tune the MLP network for clear and easy understanding. Please refer to the end of the Section 3.2. In addition, we conducted more experiments to approve the robust and consistent performance of our method. We also fixed typos.

---

### Decision · Program_Chairs · 2023-01-20

**Decision:**

Reject

**Justification For Why Not Higher Score:**

A style transfer method that combines loss terms developed in prior works for the same task. No additional insight.

**Justification For Why Not Lower Score:**

N/A

**Metareview: Summary, Strengths And Weaknesses:**

The paper proposes a style transfer method using a pretrained unconditional diffusion model. Given a content image and a style image, the paper proposes a set of loss terms to provide guidance to the diffusion model to generate images that preserve the content in the input content image but have the style depicted in the style image. It leverages loss terms from the prior work to achieve the task. For the content preservation loss terms, it uses the CUT loss, the perceptual loss, and the reconstruction loss. For the style transfer loss terms, it uses the CLIP loss (text-guided part). The results are compared to those from several prior works.

The paper receives 3 reviews. All of the reviewers consider the paper below the bar. Various shortcomings were cited. But, commonly, they consider the novelty weak. It pretty much reuses the loss terms found effective in the prior work and puts them together. No additional insights are drawn. The results are good but not overwhelming. The paper still has the contribution showing that this particular combination produces a working style transfer method. However, it may not be sufficient to meet the bar for a high-quality scientific publication.